# Steering Away from Current Amoxicillin Dose Reductions in Hospitalized Patients with Impaired Kidney Function to Avoid Subtherapeutic Drug Exposure

**DOI:** 10.3390/antibiotics11091190

**Published:** 2022-09-02

**Authors:** Cornelis Smit, Swapnoleena Sen, Elodie von Dach, Abderrahim Karmime, Pierre Lescuyer, David Tonoli, Julia Bielicki, Angela Huttner, Marc Pfister

**Affiliations:** 1Pediatric Pharmacology and Pharmacometrics Research Center, University of Basel Children’s Hospital (UKBB), 4056 Basel, Switzerland; 2Department of Clinical Pharmacy, Antonius Hospital, 8601 ZK Sneek, The Netherlands; 3Department of Pharmaceutical Sciences, University of Basel, 4056 Basel, Switzerland; 4Swiss Tropical and Public Health Institute, 4056 Basel, Switzerland; 5Center for Clinical Research, Geneva University Hospitals and Faculty of Medicine, 1205 Geneva, Switzerland; 6Division of Laboratory Medicine, Diagnostic Department, Geneva University Hospitals, 1205 Geneva, Switzerland; 7Paediatric Infectious Diseases Research Group, Institute for Infection and Immunity, St George’s, University of London, London SW17 0RE, UK; 8Division of Infectious Diseases, Geneva University Hospitals, 1205 Geneva, Switzerland; 9Certara, Princeton, NJ 08540, USA

**Keywords:** amoxicillin, pharmacokinetics, pharmacology, dose optimization

## Abstract

Current dose reductions recommended for amoxicillin in patients with impaired kidney function could lead to suboptimal treatments. In a prospective, observational study in hospitalized adults with varying kidney function treated with an IV or oral dose of amoxicillin, amoxicillin concentrations were measured in 1–2 samples on the second day of treatment. Pharmacometric modelling and simulations were performed to evaluate the probability of target attainment (PTA) for 40% of the time above MIC following standard (1000 mg q6h), reduced or increased IV dosing strategies. A total of 210 amoxicillin samples was collected from 155 patients with kidney function based on a CKD-EPI of between 12 and 165 mL/min/1.73 m^2^. Amoxicillin clearance could be well predicted with body weight and CKD-EPI. Recommended dose adjustments resulted in a clinically relevant reduction in the PTA for the nonspecies-related PK/PD breakpoint MIC of 8 mg/L (92%, 62% and 38% with a CKD-EPI of 10, 20 and 30 mL/min/1.73 m^2^, respectively, versus 100% for the standard dose). For MICs ≤ 2 mg/L, PTA > 90% was reached in these patients following both reduced and standard dose regimens. Our study showed that for amoxicillin, recommended dose reductions with impaired kidney function could lead to subtherapeutic amoxicillin concentrations in hospitalized patients, especially when targeting less susceptible pathogens.

## 1. Introduction

Given the increasing rates of antibiotic resistance and few novel antibiotics, there is growing interest in maximizing the potential of old antibiotics via modern pharmacometric tools [1]. Amoxicillin is a renally cleared, parenterally or orally administered semisynthetic beta-lactam antibiotic that has been in use for more than five decades. Despite its broad use, the current knowledge of amoxicillin dose adjustments in adult patients with various levels of kidney impairment is still limited [2]. Most studies conducted to date focused on healthy volunteers [2] or specific subpopulations, such as hemodialysis [3], critically ill [4] or pediatric [5,6] patients. EMA and FDA drug labels currently recommend dose reductions when the glomerular filtration rate (GFR) drops below 30 mL/min [7,8], whereas some clinical evidence suggests that such a reduction may expose some patients to a disproportionate risk of treatment failure [9]. This is further underlined by the observations that target attainment is already hampered for some less susceptible microorganisms when using standard dosing in patients with normal kidney function [10]. As such, a pharmacokinetic study was conducted in this population (i) to collect pharmacokinetic data in an unselected group of hospitalized patients with varying kidney function, (ii) to characterize population pharmacokinetics of amoxicillin in these patients and (iii) to perform simulations to evaluate the probability of target attainment (PTA) on the second day of treatment in patients with normal or reduced kidney function following standard and reduced IV dose regimens.

## 2. Materials and Methods

### 2.1. Patient and Data Collection

Data were available from the OPTIMAL TDM study (NCT03790631), which was a prospective, observational study conducted at the University Hospital Geneva (HUG, Geneva, Switzerland), designed to determine the role of therapeutic drug monitoring (TDM) in optimizing the dosage of several beta-lactam antibiotics, including amoxicillin, in hospitalized patients. All patients admitted to intensive care, intermediate care or hematology/transplant wards who were treated with intravenous or oral amoxicillin for a suspected or confirmed systemic bacterial infection were included. Patients with documentation of extracorporeal renal replacement therapy during treatment with amoxicillin were excluded from the analysis. For dose individualization purposes, one or two blood timed samples on the second day of treatment were drawn, preferably consisting of one midinterval and one trough level. Total amoxicillin concentrations were measured from lithium heparinized plasma samples. After protein precipitation, samples were analyzed in the multiple reaction monitoring mode on a QTRAP 4500MD hybrid triple quadrupole linear ion trap mass spectrometer (Sciex, Concord, ON, Canada) hyphenated to a UHPLC Agilent Infinity system (LC-MS/MS). The method was validated according to the Clinical & Laboratory Standards Institute (CLSI) guidelines. The study was conducted in full accordance with the Declaration of Helsinki and the principles of Good Clinical Practice. The Geneva Cantonal Ethics Commission (2018-08130) judged it to be of a quality control nature and written informed consent was waived.

For each patient, the baseline data on gender, age, body weight, height, ward and APACHEII (in the case of ICU patients [11]) and qSOFA [12] score were recorded. Serum creatinine, expressed as mcmol/L, was measured in lithium heparinate plasma as part of the routine care for each individual patient. In the intensive care unit (ICU), serum creatinine was drawn once daily at minimum. In the other patient units, serum creatinine was drawn daily throughout the acute phase of infection; once patients were stabilized, it was drawn every 1–3 days or as otherwise clinically indicated. Serum creatinine was measured using a kinetic colorimetric assay based on the Jaffé method on Roche Cobas c702 clinical chemistry analyzers using the Roche CREJ2 reagent pack. For each individual, the serum creatinine measurement closest to the first amoxicillin concentration measurement was used to estimate the glomerular filtration rate using several methods, i.e., the Cockcroft-Gault [13], CKD-EPI [14] or 4-variable MDRD [15] equations. Chronic kidney disease state was categorized using the K/DOQI criteria [16].

### 2.2. Population Pharmacokinetic Modelling 

Concentration–time data were modelled using nonlinear mixed-effects modelling using the stochastic approximation expectation–maximization (SAEM) algorithm implemented in the Monolix software package (version 2020R1 and, for final diagnostic figures, version 2021R2 [17]). Data cleaning, restructuring and visualization were performed using R (version 4.0.2 [18]) and RStudio (version 1.3.1073 [19]). One- and two-compartment models with first-order absorption, linear elimination and additive, proportional or combined error structures were fitted to the data. Interindividual variability (IIV) of the population parameters was assumed to be log-normally distributed. Nested models were compared using the −2 × log-likelihood or objective function value (OFV); part of the standard output by Monolix was based on importance sampling, where a difference of at least 3.84 points in the OFV indicated a significantly better model (*p <* 0.05), along with diagnostic criteria, such as the goodness-of-fit (GOF), parameter precision, overall model stability and conditional number. Body weight was included in the structural model scaled to a standard value of 70 kg with estimated or standard allometric scaling exponents, where allometric scaling was implemented, resulting in a significant decrease in the OFV (ΔOFV > 3.84 points, *p <* 0.05). An alternative model with estimated exponents for the weight effect did not result in a significantly better fit compared to the model with allometric scaling (ΔOFV < 3.84 points, *p >* 0.05).

Based on the random effect versus covariate plots and physiological plausibility, kidney function estimates of the CKD-EPI, MDRD or CG and other potential covariates were investigated as predictors for population pharmacokinetic parameters by implementing these in the model according to Equation (1):(1)$${\mathrm{logP}}_{\mathrm{ind}}{=\mathrm{logP}}_{\mathrm{pop}}+\beta_{\mathrm{COV}}\;\times \;\log\;\left(\frac{{\mathrm{COV}}_{\mathrm{ind}}}{{\mathrm{COV}}_{\mathrm{med}}}\right)$$
where the individual and population pharmacokinetic parameter estimates are represented by P_ind_ and P_pop_, and individual covariate values COV_ind_ are normalized with the median of the population (COV_med_) and the estimated effect of the covariate on the parameter is represented by β_COV_. In the case where covariates were significant, the estimate that resulted in the largest OFV reduction and improvement in diagnostics was included in the model.

The final model was internally validated with a prediction-corrected visual predictive check (pcVPC) and analysis of the normalized prediction distribution errors (NPDEs). Nonparametric bootstrap resampling (*n* = 1000, using the Rsmlx R package), stratified on the route of drug administration and included covariates, was performed to assess the robustness of the model and to estimate the 95% confidence intervals of the estimated PK parameters.

### 2.3. Dose Evaluations

To assess the efficacy of the target attainment and the need for amoxicillin dose adjustment in patients with varying levels of kidney function, we performed model-based dose simulations using the final PK model, including the IIV in a group of virtual patients (*n* = 16,000) with varying body weights (following a normal distribution using the median and standard deviation of the original dataset) and increasing kidney function (CKD-EPI of between 10 and 160 mL/min/1.73 m^2^ with steps of 10). Simulations were performed using Simulx as part of the Monolix suite [17]. Simulated total concentrations were transformed to free (unbound) drug concentrations by multiplying with 0.82, assuming 18% protein binding [7]. The percentage of time of free drug above the minimal inhibitory concentration on the second day of treatment (%*f*T > MIC) for different MICs was the endpoint assessed in the simulations with 40% *f*T > MIC as the primary target as proposed by the EUCAST [10,20]. The 100% *f*T > MIC was assessed as a secondary endpoint for different MICs. A 90% PTA was used as the target for the simulations for both primary and secondary endpoints. Each patient received a multiday dose regimen of amoxicillin using various dosing strategies: (1) the standard (EMA and FDA drug labels) dose of 1000 mg every 6 h (1000 mg q6h); (2) the FDA drug label recommended reduced dose of 500 mg every 12 h (500 mg q12h), recommended for patients with an impaired kidney function of <30 mL/min/1.72 m^2^; (3) the EMA drug label recommended reduced dose which was the same as the FDA label recommendation but with a loading dose of 1000 mg; (4) an alternative reduced dose with a reduced dose of 500 mg in the standard frequency of every 6 h (500 mg q6h); and (5) an increased dose of 2000 mg every 6 h (2000 mg q6h). Each amoxicillin dose was infused over 1 h, in line with the drug’s label recommendations.

## 3. Results

### 3.1. Dataset

A total of 155 patients treated with amoxicillin (with 141 (91%) IV, 9 (6%) orally and 5 (3%) both IV and orally) was included, yielding 209 amoxicillin measurements. Figure 1 shows the observed amoxicillin concentrations versus time after dose. The baseline patient characteristics are shown in Table 1, indicating a broad range of age (16–93 years) and kidney function (CKD-EPI 11.5–165 mL/min/1.73 m^2^).

### 3.2. Population Pharmacokinetic Modelling

A one-compartmental model with a combined additional and proportional residual error term and a depot absorption compartment best described the data. Interindividual error was retained for the volume of distribution (V_d_) and CL. Due to the sparsity of data in the drug absorption phase, no population value for K_a_, nor an estimate for IIV on K_a_, could be estimated with sufficient precision, as this led to high-parameter RSE values and model stability issues. Therefore, K_a_ was fixed to a value of 1.02/h, as reported in the literature [21]. To reflect the variability on K_a_ as reported in the same study [21], we additionally fixed the IIV on K_a_ at 0.2. Different fixed values for the population estimate and IIV of K_a_ were tested as a sensitivity analysis and did not lead to a better model fit (∆OFV +16.1 and +0.8, respectively, for K_a_ = 0.8 or 1.2, compared to the model with K_a_ = 1.02 and ∆OFV +0.6 and +5.0 for IIV on K_a_ = 0.1 or 0.4, respectively, compared to the model with IIV on K_a_ = 0.2). To assess the presence of flip-flop kinetics [22], the model was rerun with a large initial estimate for K_a_ (5/h), which resulted in a higher estimate of K_a_ (2.61/h), albeit again with a very large RSE value. A similar value for CL was estimated in both situations (14.9 L/h and 14.1 L/h with K_a_ fixed to 1.02 and estimated at 2.61/h, respectively) with no differences in CL between both administration routes based on the covariate plots. Altogether, this indicated that there was no evidence in our data to include flip-flop kinetics in the model. The implementation of delayed and saturable absorption kinetics with fixed values from a recent healthy volunteer study [23] did not significantly improve the model compared to the model with delayed absorption (∆OFV −3.1, *p >* 0.05) and also did not improve the goodness-of-fit. Body weight was included as a covariate for CL and V_d_ by allometric scaling, as this improved diagnostic plots, although no significant difference in the model fit was observed (∆OFV −0.8 points compared to the structural model without covariates, *p >* 0.05).

A strong influence of CKD-EPI, MDRD or CG on amoxicillin CL was visible on the covariate versus random effect parameter plots (shown for CKD-EPI in Appendix A), where kidney function estimates significantly improved the model (see Appendix A). The largest improvement of the model fit was obtained with CKD-EPI (∆OFV −141.3 compared to a model without CKD-EPI, *p <* 0.001), whereas MDRD and CG resulted in a decrease in OFV of −136.8 and −104.5 points, respectively, compared to a model without MDRD or CG (both *p <* 0.001). Deindexed CKD-EPIs, calculated by multiplying the CKD-EPI with body surface area (BSA)/1.73, did not result in a better model fit (∆OFV +20.1 compared to the model with CKD-EPI on CL, *p >* 0.05). To investigate the relevance of augmented renal drug clearance, capping of the CKD-EPI value at a maximum value was investigated. Here, capping at 120, 130 or 140 mL/min/1.73 m^2^ did not result in a significantly improved model fit compared to a model without capping of CKD-EPI (∆OFV +9.3, +7.9 or +1.6, respectively, *p <* 0.05 for all, Appendix A). Lastly, a linear function for the effect of CKD-EPI on CL, with β_CKD-EPI_ fixed to one, resulted in a comparable model fit as the model with an estimated parameter (∆OFV +1.4 points compared to the model with β_CKD-EPI_ on CL estimated (value 1.20), *p >* 0.05), and was included in the final model.

The pcVPC indicated a good agreement between the observed and simulated amoxicillin concentrations in the model (Appendix A). Good predictive performance was confirmed with the goodness-of-fit and NPDE diagnostics (Appendix A). All diagnostics were split for kidney function with no major differences in model performance in the high (CKD-EPI > 50 mL/min/1.73 m^2^) and low (CKD-EPI > 50 mL/min/1.73 m^2^) kidney function groups. Lastly, the nonparametric bootstrap resampling analysis indicated the robustness and stability of the results (Table 2), with the exception of F, which showed a rather broad 95% confidence interval. Additionally, there were some difficulties regarding the interindividual variability on the volume of distribution (ωV_d_) with a higher mean parameter value in the bootstrap analysis (0.339 versus 0.226 estimated in the final model). Model parameters of the final pharmacokinetic model are shown in Table 2.

### 3.3. Dose Evaluations

Figure 2 shows the model-based simulated PTA following IV-administered amoxicillin for the primary efficacy endpoint of 40% *f*T > MIC for different dose regimens versus kidney function with the results split for MICs. Figure 3 shows the same data split for kidney function with the MIC as the independent variable. Following the FDA label recommended reduced dose of 500 mg q12h, the observed PTA for the nonspecies-related PK/PD breakpoint MIC of 8 mg/L [20] was 92%, 62% and 38% for individuals with a CKD-EPI of 10, 20 and 30 mL/min/1.73 m^2^, respectively, being lower compared to the standard dose of 1000 mg q6h, where a PTA for an MIC of 8 mg/L of 100% was obtained in all these cases. Comparable results were obtained with the EMA label recommended reduced dose regimen, which included a loading dose (Figure 2 and Figure 3). Reasonable results were obtained with an alternative reduced regimen of 500 mg q6h, where PTAs of 100%, 96% and 85% were observed for CKD-EPI of 10, 20 and 30 mL/min/1.73 m^2^, respectively. For an MIC of 4 mg/L, a below-target PTA (<90%) with the FDA- or EMA-recommended dose reduction was seen only in individuals with a CKD-EPI of 30 mL/min/1.73 m^2^, where a >90% PTA was reached with standard dosing in this group. For lower MICs of ≤2 mg/L, the primary efficacy endpoint of 40% *f*T > MIC was reached in more than 90% of the patients in both the reduced and standard dose regimens. Notably, in individuals receiving the standard dose, the PTA for 40% *f*T > MIC of 8 mg/L dropped below 90% with kidney function of ≥ 60 mL/min/1.73 m^2^, whereas the increased dose of 2000 mg 6 h maintained a PTA > 90% up to a CKD-EPI of approximately 100 mL/min/1.73 m^2^. Sufficient concentrations were preserved across all kidney function levels for the standard dose regimen when considering MICs of ≤2 mg/L. The results for 100% *f*T > MIC are shown in Appendix A. Here, similar trends were observed, although overall, the PTA was below the clinical target of 90% in the majority of patients for all dose regimens. This included low MIC values of ≤2 mg/L (Appendix A). Appendix A shows the amoxicillin trough concentrations at the end of day two, demonstrating increased trough concentrations with lower CKD-EPI. This increase was more pronounced in the standard dose group compared to the reduced dose regimen, including the alternative reduced dose regimen of 500 mg q6h.

## 4. Discussion

We characterized the relationship between kidney function and clearance of amoxicillin and evaluated current dose reduction recommendations in a real-world setting of patients with a wide range of kidney function. Model-informed simulations revealed that patients with reduced kidney function could be at a considerable risk of undertreatment when the amoxicillin dose was reduced to 500 mg q12h according to the FDA or EMA label recommendations. This particularly applied when targeting higher MICs (≥4 mg/L) in patients with moderately reduced kidney function between 10 and 30 mL/min/1.73 m^2^. In contrast, with a standard dose of 1000 mg q6h, a PTA of close to 100% could be achieved in hospitalized patients with normal or reduced kidney function. Given the increasing rate of antibiotic resistance and low toxicity of beta-lactams and amoxicillin in particular [24], these results should spur a re-evaluation of our practice of ‘blindly’ reducing doses in renally impaired patients. We did observe increased amoxicillin concentrations with the standard (nonadjusted) dose in patients with reduced kidney function, although the clinical significance of this finding was uncertain, as specific toxicity thresholds for amoxicillin have not been established. Our findings were in line with those of other studies that also demonstrated a low target attainment with reduced doses in renally impaired patients [9]. A second study in critically ill patients did not study drug label recommended dose reductions, but similar to our results, reported adequate target attainment in renally impaired individuals following the alternative reduced dose of 500 mg q6h [4]. Such a reduction showed reasonable results in our study as well, and might be a promising alternative in patients with kidney impairment and a high risk of toxicity. Furthermore, our results showed that target attainment might be hampered in patients with normal kidney function receiving the standard dose, especially with less susceptible pathogens with MICs of approximately 4–8 mg/L. In these cases, an increased dose might be necessary, in line with earlier observations [10,25,26].

In agreement with our results, several other studies reported a correlation between amoxicillin clearance and kidney function [4,25,27]. The observed typical clearance value (17.2 L/h (95% CI 14.7–19.2 L/h) for a 70 kg individual) was similar to what was reported in earlier studies with values between 10 and 20 L/h being reported for different patient populations and healthy volunteers [2,23]. We observed a relatively high volume of distribution (56.6 L (95% CI 41.3–69.7 L) for a 70 kg individual); this was higher than that reported in the literature for mostly healthy volunteers, with values varying between 20 and 30 L [2]. Our results in this regard should be interpreted with caution due to the lack of a control group.

A strength of our study was its relatively large cohort of 155 patients compared to the populations of 12–57 individuals studied in previous population pharmacokinetic studies [4,9,25]. Moreover, we studied an unselected cohort of hospitalized patients, whereas other studies exclusively focused on a specific subpopulation [4,9]. There were some limitations that should be addressed. First, although the dataset included patients receiving both oral and IV amoxicillin, target attainment was assessed for IV amoxicillin only. Judging from the modelling results for K_a_ and the bootstrap analysis, there was insufficient power to appropriately characterize the bioavailability or absorption profile, which are known to be nonlinear for amoxicillin in high doses [23]. Most likely, this was due to the scarcity of pharmacokinetic data around the absorption phase and the low number of individuals in the oral group. In addition, our data showed no evidence directing to flip-flop kinetics. This was in contrast to one study in a small cohort of healthy volunteers receiving amoxicillin as a suspension [22]. Caution must be employed when extrapolating our IV dosing results to the oral administration of amoxicillin. Second, we only investigated serum creatinine-based kidney function estimates as a covariate for kidney function. Serum creatinine-based estimates are known to have limitations, especially in ICU patients [28]. Numerous novel diagnostic biomarkers have been identified over recent years [28,29,30,31]. These biomarkers have proven useful in predicting acute kidney injury and, in some cases, outperform serum creatinine. However, they have not yet become commonplace in clinical healthcare settings. In addition, most of these novel biomarkers have not yet been investigated thoroughly in their ability to guide drug dosing, with the exception of cystatin C [32]. Unfortunately, we did not have data for any of these biomarkers in our dataset to investigate their added value in guiding amoxicillin dosing. Future research might further enhance approaches to assess kidney function for guiding drug dosing [28]. Until then, our current body of work using the serum-creatinine-based eGFR as an indicator for amoxicillin dose adjustments could be used in clinical practice starting today. Lastly, although we included patients with a wide range of kidney function, the group with severe kidney failure (CKD-EPI < 15 mL/min/1.73 m^2^) was, with only three individuals, underrepresented in our dataset. Split diagnostics for kidney function, however, indicated a good description of amoxicillin measurements across all kidney function groups without any evidence for bias (Appendix A). Nevertheless, we do advise caution when applying these results to patients with severe kidney impairment.

## 5. Conclusions

In conclusion, the dose adjustment of IV amoxicillin according to current dose recommendations could lead to subtherapeutic drug exposure in hospitalized patients with impaired kidney function (>10 mL/min/1.73 m^2^), especially when targeting less susceptible pathogens. In this patient population, a standard amoxicillin dose of 1000 mg q6h may be appropriate to avoid the risk of treatment failure, while in renally impaired patients with a high risk of toxicity, an alternative dose of 500 mg q6h could be considered. Larger studies are warranted to assess the clinical and pharmacometric outcomes under such enhanced dosing schemes.

## Figures and Tables

**Figure 1 antibiotics-11-01190-f001:**
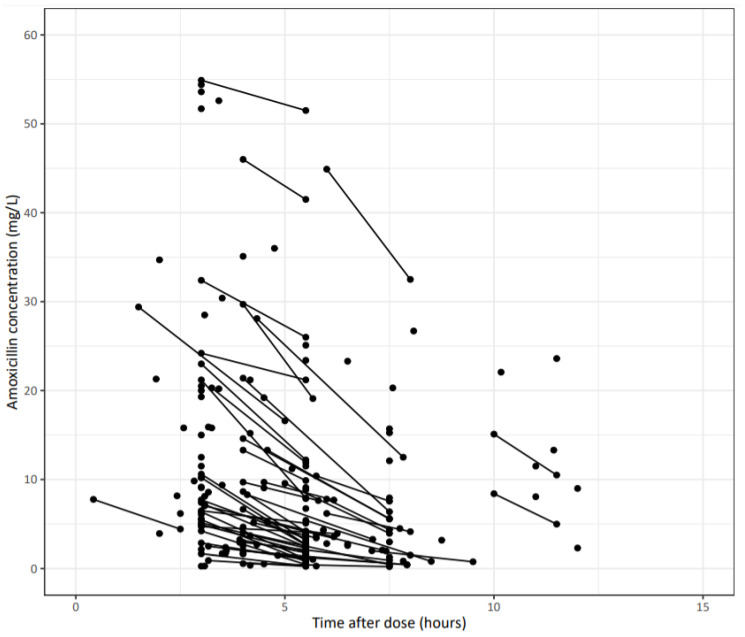
Observed amoxicillin concentrations (mg/L) versus time after the last dose (in hours). Each dot represents an observed amoxicillin concentration. Two measurements from a given patient are connected with a solid line.

**Figure 2 antibiotics-11-01190-f002:**
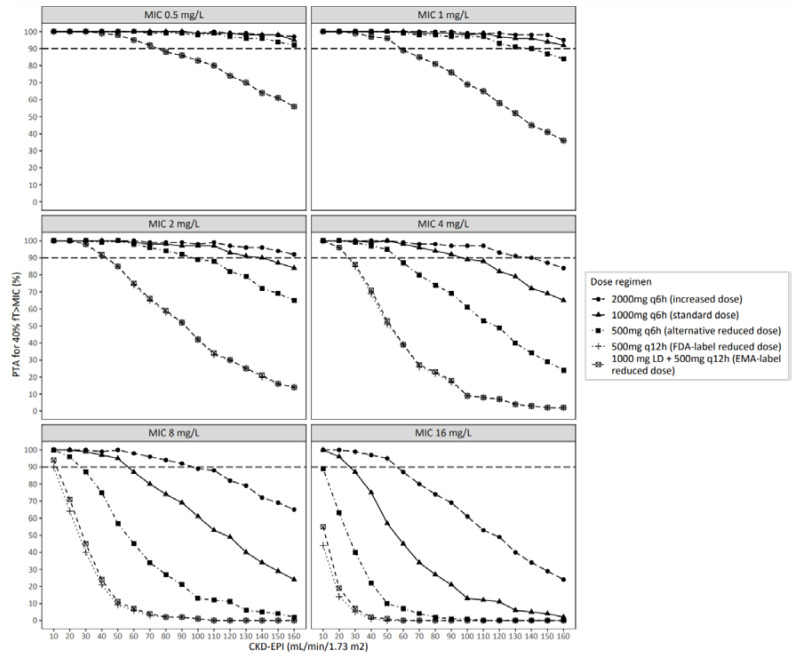
Probability of target attainment (PTA, in %) for the primary efficacy target of 40% *f*T > MIC on the second day of treatment with IV amoxicillin versus kidney function depicted with the CKD-EPI (in mL/min/1.73 m^2^) following different dose regimens (*n* = 16,000 patients per dose regimen). Results are shown for the MICs of 0.5–16 mg/L. Each line represents a different dose regimen. For reference, the dashed line shows 90% PTA as a commonly used minimum target. PTA probability of target attainment.

**Figure 3 antibiotics-11-01190-f003:**
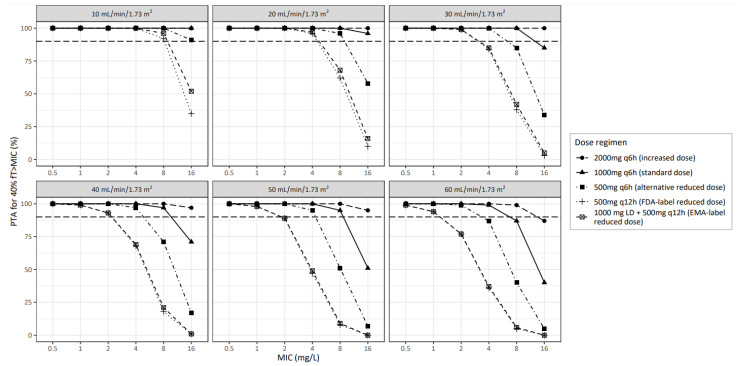
Probability of target attainment (PTA, in %) versus MIC (in mg/L) for different doses (*n* = 1000 pa-tients per combination of kidney function group and dose regimen) for the primary efficacy target of 40% *f*T > MIC on the second day of treatment with IV amoxicillin for patients with kidney function up to 60 mL/min/1.73 m^2^ based on CKD-EPI. For reference, the dashed line shows 90% PTA as a commonly used minimum target. Each line represents a different dose regimen. MIC, minimal inhibitory concentration. PTA, probability of target attainment.

**Table 1 antibiotics-11-01190-t001:** Patient characteristics of included hospitalized patients (*n* = 155).

Patient Characteristic	Value ^a^
Age (years)	72 (54–80) (16–93)
Gender (female, *n* (%))	51 (33)
Weight (kg)	70 (61–83) (30–140)
Route of administration (*n* (%))	IV: 141 (91) Oral: 9 (6) IV + oral: 5 (3)
Admitted to ICU (*n* (%))	107 (69)
Indication for amoxicillin treatment (*n* (%))	(Suspected) pneumonia: 113 (73) Wound infection: 15 (10) Catheter infection: 2 (1) Urinary tract infection: 1 (1) Other: 24 (15)
APACHE II score ^b^	26 (20–30) (3–45)
qSOFA score (*n* (%))	0: 23 (15) 1: 56 (36) 2: 55 (35) 3: 21 (14)
Serum creatinine (mcmol/L) ^d^	78 (59–106) (36–413)
Chronic kidney disease class (based on K/DOQI criteria, *n* (%)) [16] ^c^	Normal (CKD-EPI >90): 85 (54) Mild (CKD-EPI 60–89): 33 (21) Moderate (CKD-EPI 30–59): 25 (16) Severe (CKD-EPI 15–29): 9 (6) Kidney failure (CKD–EPI <15): 3 (2)
CKD-EPI (mL/min/1.73 m^2^)	94.4 (67.0–115.6) (11.5–165.8)
MDRD (mL/min/1.73 m^2^)	81.5 (54.9–110.9) (10.1–202.7)
CG (mL/min)	78.8 (44.1–108.3) (10.5–220.9)

^a^ Data shown as median (IQR) (range) unless specified otherwise; ^b^ only measured for ICU patients (*n* = 107, 1 missing value); ^c^ classified using the CKD-EPI in ml/min/1.73 m^2^; ^d^ serum creatinine was measured on the same day of the first amoxicillin concentration measurement in 67% of cases, and within 48 h in 96% of cases.

**Table 2 antibiotics-11-01190-t002:** Population pharmacokinetic parameters of the final amoxicillin model and results of the bootstrap resampling analysis (*n* = 1000).

Parameter	Estimate (%RSE)	Bootstrap Estimate (95% CI)
**Population parameters**		
F (%)	49.8 (15)	56.4 (29.6–100)
K_a_ (h^−1^)	1.02 FIX	1.02 FIX
**CL_i_ = CL_pop_ × (WT/70)^0.75^ × (CKD-EPI/94)**		
CL_pop_ (L/h)	17.2 (5)	16.9 (14.7–19.2)
V_d,i_ = V_d,pop_ × (WT/70)^0.75^		
V_d,pop_ (L)	56.6 (6)	53.6 (41.3–69.7)
**Interindividual variability**		
ωK_a_	0.2 FIX	0.2 FIX
ωCL	0.481 (8)	0.473 (0.376–0.572)
ωV_d_	0.226 (20)	0.339 (0.180–0.610)
**Residual variability**		
Additional error (mg/L)	0.146 (36)	0.206 (0.012–0.53)
Proportional error (SD)	0.226 (15)	0.175 (0.029–0.307)

CKD-EPI, chronic kidney disease epidemiology collaboration; F, bioavailability; K_a_, absorption rate constant; CL_i_, individual amoxicillin clearance; CL_pop_, population amoxicillin clearance; RSE, relative standard error; SD, standard deviation; V_d,i_, individual volume of distribution; V_d,pop_, population volume of distribution; WT, body weight.

## Data Availability

The data presented in this study are available on request from the corresponding author.

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
