# Peer review of "Steering Away from Current Amoxicillin Dose Reductions in Hospitalized Patients with Impaired Kidney Function to Avoid Subtherapeutic Drug Exposure"

_antibiotics, 2022, doi:10.3390/antibiotics11091190_

Round 1
Reviewer 1 Report
The paper is very interesting. Antibiotic therapy and dose modification in renal insufficiency is a very interesting clinical problem in ICU and in intensive nephrology. However, the topic in this paper has some biases:
1) the toxicity of antibiotic therapy cannot be assessed by the reduction of the eGFR alone. However, it is correct to speak of “ eGFR (estimated GFR)” and not of “serum creatinine”.
Recently, research has highlighted numerous biomakers that predictly indicate the eventual evolution in AKI.
In fact, when the eGFR is reduced it is late and the evolution in AKI is clear.
The dosage of the new biomarkers is also essential to modify the dosage of antibiotic therapy.
When biomarker dosage increases, AKI can still be prevented.
2) the references are not very recent, and need to be improved. I suggest citing the following references that address the topic of AKI biomarkers in Critical Care and Intensive Nephrology.
1: Jacobsen E, Sawhney S, Brazzelli M, Aucott L, Scotland G, Aceves-Martins M,
Robertson C, Imamura M, Poobalan A, Manson P, Kaye C, Boyers D. Cost-
effectiveness and value of information analysis of NephroCheck and NGAL tests
compared to standard care for the diagnosis of acute kidney injury. BMC Nephrol.
2021 Dec 1;22(1):399. doi: 10.1186/s12882-021-02610-9. PMID: 34852765; PMCID:
PMC8638090.
2: Vandenberghe W, Van Laethem L, Herck I, Peperstraete H, Schaubroeck H,
Zarbock A, Meersch M, Dhondt A, Delanghe S, Vanmassenhove J, De Waele JJ, Hoste
EAJ. Prediction of cardiac surgery associated - acute kidney injury (CSA-AKI) by
healthcare professionals and urine cell cycle arrest AKI biomarkers
[TIMP-2]*[IGFBP7]: A single center prospective study (the PREDICTAKI trial). J
Crit Care. 2022 Feb;67:108-117. doi: 10.1016/j.jcrc.2021.10.015. Epub 2021 Nov
3. PMID: 34741963.
3: Yang HS, Hur M, Lee KR, Kim H, Kim HY, Kim JW, Chua MT, Kuan WS, Chua HR,
Kitiyakara C, Phattharapornjaroen P, Chittamma A, Werayachankul T, Anandh U,
Herath S, Endre Z, Horvath AR, Antonini P, Di Somma S; GREAT Network. Biomarker
Rule-in or Rule-out in Patients With Acute Diseases for Validation of Acute
Kidney Injury in the Emergency Department (BRAVA): A Multicenter Study
Evaluating Urinary TIMP-2/IGFBP7. Ann Lab Med. 2022 Mar 1;42(2):178-187. doi:
10.3343/alm.2022.42.2.178. PMID: 34635611; PMCID: PMC8548247.
4: Jia L, Sheng X, Zamperetti A, Xie Y, Corradi V, Chandel S, De Cal M, Montin
DP, Caprara C, Ronco C. Combination of biomarker with clinical risk factors for
prediction of severe acute kidney injury in critically ill patients. BMC
Nephrol. 2020 Dec 10;21(1):540. doi: 10.1186/s12882-020-02202-z. PMID: 33302892;
PMCID: PMC7731753.
5: Naorungroj T, Serpa Neto A, Yanase F, Bittar I, Eastwood GM, Bellomo R.
NephroCheck® Quality Test. Blood Purif. 2021;50(4-5):489-491. doi:
10.1159/000511727. Epub 2020 Dec 8. PMID: 33291111.
6: Chimenz R, Chirico V, Basile P, Carcione A, Conti G, Monardo P, Lacquaniti A.
HMGB-1 and TGFβ-1 highlight immuno-inflammatory and fibrotic processes before
proteinuria onset in pediatric patients with Alport syndrome. J Nephrol. 2021
Dec;34(6):1915-1924. doi: 10.1007/s40620-021-01015-z. Epub 2021 Mar 24. PMID:
33761123.
7: Monardo P, Lacquaniti A, Campo S, Bucca M, Casuscelli di Tocco T, Rovito S,
Ragusa A, Santoro A. Updates on hemodialysis techniques with a common
denominator: The personalization of the dialytic therapy. Semin Dial. 2021
May;34(3):183-195. doi: 10.1111/sdi.12956. Epub 2021 Feb 16. PMID: 33592133.
However, the paper is very interesting and can be a "work in progress" for interdisciplinary collaborative study.
Reviewer 2 Report
This manuscript describes the development of a population pharmacokinetic model using data from patients with renal impairment. Simulations were performed in order to the probability of target attainment with various dosing schemes assuming various levels of renal impairment.
The manuscript is clearly written. However, there are methodological aspects that have to be solved, which can significantly alter the result of the study.
Introduction
1. Please remove the sentence: “with few pharmacometric studies published to date”. There are some other publications where the effect of renal clearance was explored for dose optimization using population pharmacokinetics. Please cite the previous work that has been done and highlight the most important findings.
Please dig in the literature to make sure that all the area is covered. Some of the studies that have to be mentioned are:
https://www.ajkd.org/article/S0272-6386(21)00002-0/fulltext
https://academic.oup.com/jac/article/68/11/2600/833044?login=true
https://journals.asm.org/doi/full/10.1128/AAC.01368-15
Materials and Methods
2. Materials and Methods: Patients and Data collection. The samples obtained from the subjects were: “For dose individualisation purposes, one or two blood samples on the second day of treatment were drawn, preferably consisting of one mid-interval and one trough level timed.”. Assuming that all the samples were obtained from these time points it is impossible to characterize accurately the absorption phase and probably the clearance too. This is also proved by the sparseness of data in Figure 1. In the cases where only this type of data is available then Bayesian estiation using a previously developed model with more data points should be used.
3. Based on a previous publication were enough points were collected to characterize the absorption phase nonlinear absorption was noted : https://academic.oup.com/jac/article/71/10/2909/2388123?login=true . Based on this observations it was shown that no point in high-dosing regimens. How was this phenomenon accounted in your model? Knowing that amoxicillin displays fli-flop phenomenon (https://www.ncbi.nlm.nih.gov/pmc/articles/PMC4536115/ ), i.e. absorption drives clearance … mischaracterization of the absorption phase by the model would lead to mischaracterization of clearance as well.
4. Except for “Kidney function estimates CKD-EPI, MDRD or CG” which other covariates where explored for clearance in you rmodel ? Based on the range of values shown in Table 1 age should also be tested on clearance and volume of distribution.
Results
5. Please change the title from “ Pharmacometric modelling” to Population pharmacokinetic modeling. Pharmacometrics is a very large term that includes population modeling but many other approaches as well.
6. The problems in the methodology followed and data collection are evident in the results as well. Please re-develop your model using Bayesian Estimation using a published model that was developed with an adequate sample size and using adequate sampling.
7. Table 2 The 95% CI of the bootstrap estimates for bioavailability provide evidence that the fraction absorbed in the central compartment (or biovailability) is no accurately estimated which makes sense as the absorption phase is not well characterized.
8. In addition, the population parameter of clearance is underestimated as based on non-compartmental studies as well as other population pk studies amoxicillin’s clearance is ~ 25 L/h. The difference is the estimated clearance is very significant for your study as a value of 25 would significantly diversify your findings.
9. In addition please note that the SAEM algorithm does not provide an objective function value. The OFV that monolix reports is either form importance sampling or from linearization. I strongly suggest to use the importance sampling method and to compare the BIC that monolix provides. https://monolix.lixoft.com/tasks/log-likelihood-estimation/
Discussion
10. The difference noted in the volume of distribution as mentioned in the discussion is because absorption and mainly clearance are not well estimated. Thus, the model to compensate for the underestimated clearance increased the volume of distribution to match the observed values.
Supplementary materials
11. Figure S1: Add regression line and r-squared
12. Figure S2: Please present the pots in natural and non-log scale and add the least-squares regression curve preferably in red. In addition add the IWRES or CWRES versus time and versus predicitons
13. Figure S3: Add the percentiles of the predicted NPDE and correct the figure legend.
14. Figure S4:Add the predicted percentiles.
Round 2
Reviewer 1 Report
The authors may not have guessed at the reviewer's remarks. In the nephrology / internal field, creatinine cannot be the ideal biomarker to evaluate the eGFR. Perhaps in clinical practice it could be, but when designing a study for publication purposes, the premises must be different.
The corrections are by no means adequate and are very superficial. The suggested references are only inserted, but not commented and have not been the origin of an important modification of the "discussion".
Therefore, the paper does not have the minimum characteristics to be published on Antibiotics.
However, the authors are advised to re-evaluate the study design and pay more attention to an important topic in nephrology / internal field such as new markers of renal function.
Author Response
We respectully disagree with the reviewer regarding the use of serum creatinine based measurements in a study for publication purposes. Our primary focus in the current paper is to investigate the necessity of dose reductions with varying kidney function. Since we want our results and conclusions to be able to be directly implemented in routine clinical care, it is key to use covariates in the analysis that are also available in commonday practice. Therefore, we have chosen to investigate serum creatinine based kidney function estimates in this study. However, we do acknowledge the limitations of serum creatinine as we have discussed this as a limitation in the paper. We unfortunately do not have any measurements of kidney biomarkers available in our dataset and therfore cannot include these in our analysis. Nevertheless, we have shown that using serum creatinine biomarkers, we can adequately predict amoxicillin clearance. Our analysis results in practical dose recommendations based on serum creatinine that can be used by clinicians worldwide today. In the revised version of the manuscript we have added some more in-depth discussion on novel kidney biomarkers in the section where we mention the limitations of serum creatinine based kidney function estimates (lines 312-324). We have included the fact that we do not have such data available, but state that future research should focus on validating these novel biomarkers in guiding drug dosing.
Reviewer 2 Report
The authors have provided detailed explanations to the comments and have significantly improved their manuscript.
Author Response
We thank the reviewer for these comments